# Leaf Aquaporin Expression in Grafted Plants and the Influence of Genotypes and Scion/Rootstock Combinations on Stomatal Behavior in Grapevines Under Water Deficit

**DOI:** 10.3390/plants13233427

**Published:** 2024-12-06

**Authors:** Andrea Galaz, Alonso G. Pérez-Donoso, Marina Gambardella

**Affiliations:** Facultad de Agronomía y Sistemas Naturales, Pontificia Universidad Católica de Chile, Santiago 7820436, Chile; apgalaz@uc.cl

**Keywords:** water stress, drought resistance, anisohydric behavior, stomatal conductance, Vitis aquaporins

## Abstract

This study investigates the impact of water stress on grapevines, specifically examining the role of rootstocks and aquaporins. Two experiments on potted plants were conducted in central Chile during the summer, under conditions of high water demand, involving various rootstock genotypes and combinations of Cabernet Sauvignon (CS) grafted onto rootstocks. Significant differences were observed among plants in terms of stem water potential, stomatal conductance, and growth rate. Notably, the CS/CS combination consistently displayed the slowest growth rate, regardless of the irrigation treatment. The study also analyzed the expression levels of plasma membrane intrinsic protein (PIP) and tonoplast intrinsic protein (TIP) aquaporins in the leaves of grafted plants. Specifically, *VvPIP2;2* aquaporins showed reduced expression after 14 days without irrigation, whereas *VvTIP1;1* and *VvTIP2;1* expression levels correlated positively with g_s_ responses in grafted plants, suggesting their role in modulating water content in leaves under water stress. TIP aquaporins likely play a significant role in the differential responses of CS plants towards near-isohydric or anisohydric behavior. The CS/CS combination exhibited near-isohydric behavior, correlating with lower TIP aquaporin expression, while the combination of CS onto 1103P and 101-14 showed higher expression, indicating anisohydric behavior. The findings suggest that grafted plants are more resilient to water stress, supporting the idea that rootstocks can mitigate the effects of water stress on the scion.

## 1. Introduction

Currently, grapevine cultivation worldwide is facing significant challenges due to reduced winter rainfall and higher summer temperatures, as noted by the IPCC [1]. This situation underscores the pressing need for efficient water management to sustain adequate wine grape production.

One approach to mitigate water scarcity is the adoption of rootstocks. Initially introduced in response to the devastating phylloxera (*Daktulosphaira vitifoliae*) infestation in the late 19th century, which nearly wiped out European vineyards, rootstocks offer more than just phylloxera resistance. They also bring benefits such as regulating cultivar vigor [2], enhancing nutrient uptake [3], and increasing drought tolerance [4]. Combining scion and rootstock varieties that can endure prolonged drought periods without compromising growth or wine quality represents a promising strategy under investigation [5,6]. However, additional research is needed to comprehensively understand the distinct physiological changes triggered by different rootstock types and to assess their diverse responses compared to grapevines grown on their own roots (own-rooted grapevines).

Various studies have investigated the stomatal behavior of non-grafted cultivars concerning water use efficiency [7,8,9]. According to these studies and in simplified form, cultivars are categorized into two types based on their stomatal response to water stress. Isohydric cultivars react to xylem tension induced by high water demand and water scarcity in the soil by closing stomata to prevent water loss [8,10]. However, this conservation mechanism, while reducing transpiration, also lowers CO_2_ assimilation, thereby impacting plant growth and productivity. Conversely, anisohydric cultivars exhibit less responsiveness to water stress, keeping stomata open, which minimally affects their growth rate but elevates the risk of xylem cavitation when soil water availability fails to meet water demand [8,11]. Furthermore, in this case, prolonged water scarcity may lead to leaf shedding, significantly compromising production [11]. However, this classification is often ambiguous and varies depending on environmental factors [12], soil characteristics [13], duration of water stress [14], and the specific combinations of scion and rootstock [15,16]. This ambiguity is highlighted in studies by Hochberg et al. [17] and Tramontini et al. [18], where Cabernet Sauvignon is classified as isohydric, while Lovisolo et al. [12] regard it as an anisohydric cultivar. Therefore, it should be considered a complex trait that should be studied in each genotype, since it is of great interest for vineyard management.

Another plant mechanism to cope with water stress involves the regulation and functionality of aquaporins, which are transmembrane water channel proteins facilitating water movement [19,20]. These proteins are expressed in tissues where water transport from roots to leaves occurs. In *Vitis vinifera* L., plasma membrane intrinsic proteins (PIPs), identified as aquaporins, have been extensively studied. It has been observed that they exhibit diurnal responses to various environmental cues [19,20,21] and facilitate water transport in both roots and leaves [22]. Another category of aquaporins, tonoplast intrinsic proteins (TIPs), may promote water movement from the vacuole to the cytosol under conditions of high water demand or function as cytoplasmic osmoregulators by aiding in the movement of ions and small solutes from the vacuole to the cytosol during stress recovery in leaves and roots [22,23,24].

The expression levels of aquaporins in leaves vary according to water stress conditions, with their response influenced by stomatal conductance depending on the expression site (plasma membrane or tonoplast) [20]. Therefore, if the expression levels of TIP aquaporins are maintained or increased under high water demand conditions, they may lead to near-anisohydric behavior by sustaining water flow into tissues [21,25].

The impact of rootstock on stomatal behavior and the involvement of aquaporins in how grafted grapevines react to water stress have been overlooked in prior studies. Previous research indicates that the physiological responses of grafted cultivars to water scarcity could be shaped by the characteristics of the rootstock [26,27,28].

Based on this understanding, this study aimed to evaluate and compare the stomatal conductance responses and growth rates across various grapevine genotypes, including both own-rooted cultivars and different graft combinations of Cabernet Sauvignon (CS) grafted on three different rootstocks, under water deficit conditions. It also investigated the expression levels of PIP and TIP aquaporins in the leaves of grafted grapevines to understand how different rootstocks influence aquaporin responses to water scarcity. Additionally, the research sought to determine the role of rootstocks in mitigating the effects of water stress on the scion, focusing on the differential responses of grapevine genotypes towards isohydric and anisohydric behaviors. Finally, the study aimed to identify specific aquaporins, such as *VvPIP2;2*, *VvTIP1;1*, and *VvTIP2;1*, that play significant roles in modulating water content in leaves under water stress conditions, and correlate their expression levels with stomatal responses in grafted plants.

## 2. Results

This study comprised two distinct experiments. The first experiment focused on four non-grafted genotypes, including two cultivars (Carménère and Cabernet Sauvignon, CAR and CS, respectively) and two rootstocks (1103P, known for its drought tolerance, and SO4, characterized by its low drought tolerance). The second experiment investigated scion/rootstock combinations, totaling three combinations. In both experiments, plants underwent a progressive water deficit phase (non-irrigation, Ni), followed by a recovery phase (Rec). These plants were contrasted with plants under full irrigation (Fi). Additionally, in the second experiment, the expression levels of aquaporins were analyzed after both the water stress and recovery phases. In the experiment concerning scion/rootstock combinations, a single bud of the CS cultivar was omega-grafted onto a 20 cm rootstock stem [29].

The experiments took place during the summer months of January and February in central Chile, reflecting the typical growth conditions for this species. The vapor pressure deficit (VPD) recorded during measurements exceeded 3.81 kPa, indicating that plants experienced stressful environmental conditions conducive to high water demand and the onset of water stress. A detailed table outlining the environmental conditions at the evaluation site is provided as Appendix A.

### 2.1. Stem Water Potential and Stomatal Conductance

Figure 1A presents the stem water potential (Ψ_stem_) values for the first experiment on own-rooted genotypes subjected to water stress treatments. On the first day of evaluation, no significant differences were observed among genotypes, and after 7 days of water stress, differences were only associated with irrigation treatment (*p* = 0.0056). After 14 days of water stress, significant differences were detected between water treatments (*p* = 0.01716) and among genotypes independent of water treatment (*p* = 0.04073), but no interaction between them. Specifically, the SO4 genotype exhibited the lowest Ψ_stem_ value (−1.75 MPa), while the CAR genotype experienced the less negative Ψ_stem_ (−1.24 MPa). The CS and 1103P genotypes displayed intermediate responses. Following a week under the recovery irrigation treatment, plants exhibited Ψ_stem_ magnitudes similar to those observed at the beginning of the experiment.

At the beginning (day 1), stomatal conductance (g_s_) values were not different between genotypes, but after 7 days of water stress (Figure 1B) they exhibited differences attributed to water treatment (*p* = 0.00032) and genotypes (*p* = 0.0072), but no interaction was observed. Notably, the 1103P rootstock consistently demonstrated higher g_s_ levels regardless of irrigation treatment, while the SO4 genotype experienced the most substantial decrease in g_s_ due to water shortage. Meanwhile, the CAR and CS genotypes displayed intermediate responses. In contrast to observations regarding Ψ_stem_, no genotype-related differences were observed after 14 days without irrigation, with all plants reaching minimum g_s_ values. Following one week of recovery, plants showed increased g_s_ levels, although they did not reach the levels observed in control plants, and no distinctions among genotypes were evident.

Figure 2A depicts the stem water potential (Ψ_stem_) for the examined scion/rootstock combinations in the second experiment. Initially, the CS/1103P combination exhibited a significantly lower Ψ_stem_ compared to other combinations (*p* = 0.0063). However, after 7 and 14 days of non-irrigated treatment, no differences were observed among combinations, with distinctions arising solely due to irrigation treatment (*p* = 0.034 and *p* = 0.0049, respectively). Throughout the trial, the CS/101-14 combination consistently displayed the lowest Ψ_stem_ values under the Ni treatment. Stomatal conductance (g_s_) did not exhibit significant differences among combinations on any evaluation day (Figure 2B). All significant differences observed were attributed to irrigation treatment, starting on day 7 (*p* = 0.0012), a trend that persisted until the conclusion of treatment (*p* = 0.0003). Following one week of recovery, all combinations showed signs of recovery, although they were unable to reach initial levels, thereby maintaining differences between irrigated and non-irrigated combinations (*p* = 0.0339).

When examining the relationship between Ψ_stem_ and g_s_ (Figure 3A), differences statistically significant among genotypes can be observed (*p* = 0.0083). The genotype 1103P showed a smaller decrease in g_s_, which could suggest an anisohydric response. In the case of the CS, CAR, and SO4 genotypes, for the same water potential range, lower g_s_ values were observed, suggesting a greater sensitivity to water stress. That could be associated with near-isohydric behavior (Figure 1).

In the scion/rootstock combination experiment, the relationship between Ψ_stem_ and g_s_ (Figure 3B) showed more attenuated differences for generally lower g_s_ magnitudes. In this case it is not possible to distinguish a stress response pattern.

### 2.2. Above-Ground Growth Rate

Figure 4A displays the shoot growth rate for the four own-rooted genotypes studied. While no significant differences were observed among genotypes initially, Ni plants exhibited decreased growth rates or ceased growth altogether as water stress progressed, resulting in significant differences due to water treatment (*p* = 0.01305).

In Figure 4B, the shoot growth rate for scion/rootstock combinations is depicted. During the first week of evaluations (days 1–7), no differences were noted between irrigated and non-irrigated plants. However, the CS/CS combination consistently displayed the slowest growth rate, regardless of irrigation treatment (*p* = 0.0158). By the final week of water stress treatment (days 7–14), an interaction between irrigation treatment and combinations emerged (*p* = 0.003). The CS/101-14 Fi combination maintained the highest growth rate, while CS/CS Fi and all non-irrigated combinations exhibited the lowest growth rates. Following one week of recovery, all combinations demonstrated similar growth rates, with no discernible differences between treatments.

### 2.3. Aquaporin Activity

In experiment two, scion/rootstock combinations, the expression of PIP and TIP aquaporins was measured in leaves at two time points; at 14 days Ni and after one week of recovery. Figure 5 depicts the expression levels of PIP aquaporins. Regarding *VvPIP2;1* (Figure 5A), no differences were observed between combinations or water treatments after 14 days of treatment. For *VvPIP2;2* (Figure 5C), expression levels were low in both water treatments, with slightly higher levels in Fi plants, although these differences were not statistically significant (*p* = 0.061).

After one week of recovery, no differences in expression levels due to water treatment were observed for the studied PIP aquaporin genes (Figure 5B,D). However, significant differences were noted in the expression levels of *VvPIP2;1* and *VvPIP2;2* (Figure 5B,D, respectively) due to the grafting combinations (*p* = 0.005 and *p* = 0.014, respectively), with CS/101-14 exhibiting higher expression levels of both aquaporins.

Figure 6A,C shows the expression levels of *VvTIP1;1* and *VvTIP2;1* aquaporin genes. In both cases, differences are observed between Ni and Fi treatments, with higher expression in Fi (*p* = 0.0067 and *p* = 0.0280, respectively).

In plants undergoing recovery, expression levels of the *VvTIP1;1* aquaporin gene (Figure 6B) increased notably. Interestingly, differences were attributed to the genotype combinations (*p* = 0.014), particularly with CS/CS displaying lower expression levels than other combinations, which exhibited uniformly high expression levels without notable differences among them. Regarding *VvTIP2;1* (Figure 6D) aquaporin expression levels, differences were also observed due to combinations (*p* = 0.0004), with CS/CS exhibiting the lowest expression levels. No significant differences were noted between control (Fi) and recovered plants (*VvTIP1;1 p* = 0.227, *VvTIP2;1 p* = 0.768).

Considering the differences in the expression of *VvTIP2;1* aquaporins between scion/rootstock combinations (Figure 6B), a correlation analysis was performed between these values and stomatal conductance. As shown in Figure 7A, considering Fi and Ni plants at 14 days, a high correlation was obtained (0.7412), clearly indicating that the expression of this gene is related to stomatal behavior. For the different scion/rootstock combinations in Fi and Rec conditions, the correlation was not significant, since in both cases the plants had an adequate water supply. It has been suggested that TIP aquaporins may regulate leaf water potential, contributing to differences between isohydric and anisohydric plants [25].

## 3. Discussion

In the initial experiment, the stem water potential and stomatal conductance values allowed us to observe distinct behaviors among the genotypes studied under water stress conditions. As described in the literature, certain genotypes tend to maintain stem water potential through a rapid stomatal closure response, resulting in reduced g_s_ values and prevention of dehydration [8,10]. Conversely, other genotypes keep their stomata open for longer periods, leading to a decrease in Ψ_stem_ to very negative values, potentially causing cavitation and plant mortality.

In this study, plants were subjected to severe progressive water stress conditions. After 14 days without irrigation, all genotypes showed a decrease in g_s_ to approximately 20 mmol m^−2^ s^−1^, indicating extreme stomatal closure induced by water stress, consistent with previous research [10,14,30]. Additionally, the Ψ_stem_ after 14 days reached values of −2.2 MPa. However, after 7 days without irrigation, the 1103P rootstock exhibited the highest g_s_, along with a moderate reduction in Ψ_stem_. This indicates that despite the stress conditions, it kept its stomata open and showed rapid recovery, aligning with previous descriptions of its drought resistance. In contrast, the SO4 rootstock showed the lowest g_s_ values and slower recovery compared to CS, consistent with results obtained by Bondada and Shutthanandan [31].

Furthermore, irrigation treatments significantly influenced growth rates, with 1103P and SO4 being the least affected during water scarcity periods. Despite lower Ψ_stem_ and g_s_ values, the growth rate remained stable in the 1103P rootstock. However, this was not the case for CS and CAR, as both experienced a decline in growth rate.

Although definitive conclusions cannot be drawn from this experiment, it appears that the 1103P rootstock shows a greater ability to keep stomata open under progressive stress conditions without significantly reducing its water potential. This behavior aligns with an anisohydric response. Conversely, the other three genotypes studied seem to react with more pronounced stomatal closure, approaching near-isohydric behavior, although CS has been previously classified as anisohydric [16,17,32]. It should be noted that the differences between isohydric and anisohydric behaviors are more evident under moderate water stress conditions [13], so the differences in this experiment were only noticeable after 7 days without irrigation.

In the second experiment, no significant differences were observed in Ψ_stem_ and g_s_ between scion/rootstock combinations throughout the evaluation period (Figure 2). The parameters in this trial exhibited similar behavior to those observed in the first trial; however, the minimum levels of Ψ_stem_ and g_s_ recorded after 14 days were higher than in the first trial. This is likely due to more moderate environmental conditions during the trial period (Appendix A); however, it is not excluded that the difference between the experiments is due to the ability of the rootstocks to mitigate the effect of water deficit. The differences found are attributed solely to irrigation treatments. The CS/CS and CS/1103P combinations gradually decreased their Ψ_stem_ to approximately −1.5 MPa, indicating moderate water stress. In contrast, the CS/101-14 combination decreased its Ψ_stem_ to approximately −1.7 MPa (Figure 2A). After a week of recovery, these parameters improved in all combinations, with water potential reaching values similar to fully irrigated (Fi) plants.

Irrigation treatments significantly influenced growth rates, with the CS/1103 and CS/101-14 combinations being the least affected. In contrast, CS/CS had a very low growth rate even under the fully irrigated treatment. This supports previous research suggesting that rootstocks can mitigate the scion’s response to water stress by affecting soil water extraction and regulating stomatal sensitivity [2,16,26,33,34]. Although this aspect could not be verified in this study, the growth rate results could reinforce this hypothesis.

Another mechanism involved in the stress response is related to aquaporin expression. TIP aquaporins facilitate the movement of water and solutes during cellular homeostasis, particularly affecting water flow from the vacuole to the cytoplasm during rehydration [22,24]. On the other hand, PIP aquaporins have been associated with improved hydraulic conductance in leaves, which is crucial for maintaining water transport efficiency under water deficit conditions [35]. Previous studies have shown that in response to water stress, aquaporin expression decreases, while it rises again during recovery [20,23,24,25,36,37].

In this study, the expression of two PIP aquaporins (*VvPIP2;1* and *VvPIP2;2*) and two TIP aquaporins (*VvTIP1;1* and *VvTIP2;1*) was analyzed. It should be noted that the expression levels of these aquaporins were only obtained in the second scion/rootstock combination trial. Expression levels were obtained from leaves of plants subjected to fully irrigated (Fi) and non-irrigated (Ni) treatments on day 14, and then at the end of the recovery period.

The results were unclear for *VvPIP2;1* and *VvPIP2;2* as no reduction in their expression was observed in any of the cases, showing a statistically non-significant trend. Similar response patterns were observed in Tempranillo leaves grafted on different rootstocks subjected to water stress, since *VvPIP2;1* and *VvPIP2;2* only showed a response in those samples grafted on 1103P, but no response in samples obtained from rootstocks R110 and 161-49C [38]. Generally, expression levels were lower in stressed plants from the Ni and Rec treatments, except for the CS/1103P combination, where *VvPIP2;1* expression was higher in the Ni treatment after 14 days.

On the other hand, *VvTIP1;1* and *VvTIP2;1* aquaporins showed a significant reduction in their expression levels in Ni plants compared to Fi plants (Figure 6A,B, *p* = 0.0067 and *p* = 0.0280, respectively). After the recovery period, no statistical differences were observed between water treatments. However, *VvTIP2;1* expression remained at similar or lower levels in the recovery (Rec) treatment compared to Fi plants.

After the recovery period, our results showed higher expression levels for the four aquaporin genes analyzed. Significant differences were observed between scion/rootstock combinations (Figure 5B,D and Figure 6B,D). In particular, CS plants grafted onto 1103P and 101-14 rootstocks showed higher expression levels compared to homografts (CS/CS). Although the relative expression levels of *VvTIP1;1* were lower than those of the other AQP genes, Rec plants showed higher mean expression of *VvTIP1;1*, suggesting a potential role for this AQP in facilitating faster leaf hydration recovery.

Previous research on the Chasselas cultivar revealed that reduced expression levels of AQPs *VvTIP1;1* and *VvPIP2;1* were associated with decreased petiole hydraulic conductivity under water stress, suggesting their role in regulating water transport to the leaves [39]. Another study on Chardonnay showed that during stress acclimation, the foliar expression of aquaporins *VvPIP2;1*, *VvTIP1;1*, and *VvTIP2;1* decreased, with *VvPIP2;2* being undetectable in the leaves at this specific time [35]. These results closely reflect our findings in non-irrigated combinations, emphasizing the importance of aquaporins in regulating water movement under variable environmental conditions.

At the end of the 14-day irrigation experiment, a positive correlation was observed between the expression levels of *VvTIP2;1* and stomatal conductance (g_s_) in grafted plants. This relationship was less evident for *VvTIP1;1*. This finding reinforces the role of *VvTIP2;1* in regulating leaf water content under stress conditions, as supported by previous studies [25,35,39,40]. Specifically, the positive relationship indicates that aquaporin expression decreases under water stress conditions, which aligns with the low growth rates and g_s_ values observed in Ni plants. Thus, *VvTIP2;1* expression may be more closely related to the initial response to water deficit and the drastic reduction in g_s_ to the minimum levels observed in all combinations on day 14, at the end of the stress period. During the recovery period, no clear relationship was observed between *VvTIP2;1* expression and g_s_, possibly due to the faster recovery of *VvTIP2;1* expression compared to g_s_.

Previous experiments in grapevine and bean (*Phaseolus vulgaris* L.) confirm that TIP aquaporin expression decreases within 48 h under water stress, with slower recovery once water supply is restored [23,35,41]. In plants subjected to prolonged water stress, *VvTIP1;1* reached lower levels but showed higher expression after 7 days of recovery [23,25], similar to the results observed in this study in CS/1103P.

TIP aquaporins regulate cellular osmoregulation, influencing the differences between isohydric and anisohydric plants [23,25]. The decrease in aquaporin expression at sites of water loss, such as leaves, involves complex trafficking systems to export large amounts of TIPs and reduce tonoplast permeability. In this study, the CS/CS combination exhibited near-isohydric behavior, correlated with lower expression of PIP and TIP aquaporins during recovery, while Cabernet Sauvignon grafted onto 1103P and 101-14 showed higher expression linked to anisohydric behavior. This differential response of CS when grafted onto different rootstocks supports the argument that the binary classification of plant water use strategies into isohydric and anisohydric is an oversimplification. Lavoie-Lamoureux et al. [42] presented a range of stomatal sensitivities in grapevines, influenced by the interaction between genotypes and soil environment. Herrera et al. [43] further showed that grapevine stomatal responses to low water potentials increase as the growing season progresses, indicating a dynamic water use strategy. Hochberg et al. [44] argued that it is a plant–environment interaction, with the same plant capable of exhibiting both isohydric and anisohydric behaviors depending on environmental conditions. Therefore, the plant’s water use strategy is not a fixed trait but a dynamic interaction with the environment, requiring more refined methods for precise characterization. In this context, the differential expression and activity of TIP aquaporins may be one of the mechanisms explaining the dynamic spectrum of water use exhibited by plants.

## 4. Materials and Methods

### 4.1. Plant Material and Growing Conditions

In the first experiment, four ungrafted genotypes were evaluated. The rootstocks 1103P (*V. berlandieri* × *V. rupestris*) and SO4 (*V. berlandieri* × *V. riparia*) were described as drought-tolerant and low drought-tolerant, respectively [11]. Additionally, the cultivars Carménère and Cabernet Sauvignon were included. Plant material was collected during the winter (July) from the cultivar collection at the experimental station of the Pontificia Universidad Católica de Chile, located in Pirque (33°40′12.35″ S, 70°35′06.96″ W), and rooted in a heated propagation bed in a greenhouse.

In the experiment involving scion/rootstock combinations, a single bud of the Cabernet Sauvignon cultivar was omega-grafted onto a 20 cm rootstock stem [29]. The scion/rootstock combinations were as follows: CS/CS, CS/1103P, and CS/101-14.

All plants were grown in 3 L pots filled with a substrate composed of sand, peat, and perlite in a 2:1:1 volume ratio. Once established, each plant received 10 mg of slow-release fertilizer (Basacote^®^, Compo Expert GmbH, Münster, Germany). When the plants reached a height of 20 cm, they were watered weekly with 50 mL of a 25% Hoagland solution (Modified Hoagland Basal Salt Mixture H353, Phytotech Labs, Lenexa, KS, USA). To promote uniform growth, only one vertical shoot was allowed to develop, which was supported with a stake. Lateral shoots or clusters were systematically removed. The plants were initially grown under greenhouse conditions until December. Subsequently, they were moved outdoors and placed under a white mesh. Watering was carried out every two days, ensuring that the pots were watered to their maximum capacity (−0.033 MPa), maintaining adequate soil moisture levels until the start of the experiments.

In the initial experiment, conducted between 23 January and 13 February 2017 (DOY 23–44), 32 self-rooted plants were used. The second experiment, consisting of 36 plants, focused on scion/rootstock combinations and took place between 20 February and 13 March 2017 (DOY 51–72).

In both experiments, two water treatments were used. The first treatment, called full irrigation (Fi), involved watering the plants daily to replace 100% of the water lost through transpiration for 21 days, as estimated by gravimetry. The second treatment subjected the plants to a 14-day period without irrigation (Ni), and once this period was completed, the plants were subjected to a 7-day recovery treatment (Rec). At the beginning of the Rec period, the plants were watered until saturation, and subsequently the water was replenished daily in the same way as in the Fi treatment. Throughout these experiments, the pots were protected with aluminum foil to prevent evaporation from the substrate surface.

### 4.2. Stem Water Potential, Stomatal Conductance, and Environmental Conditions

In both experiments, water relations measurements were taken at four specific time points: day 1 (the first day of treatment, following saturation watering of all plants the previous day), day 7, day 14, and day 21 (during the recovery period). These measurements were taken during the period of the day with the highest water demand, typically between 11:00 and 15:00 h. Soil water content was determined using a capacitance probe (FDR Sensor GS-1^®^ from Decagon Devices Inc., Pullman, WA, USA) inserted into the pot in the root zone.

Stem water potential (Ψ_stem_) was measured on a fully expanded leaf located between nodes 7 and 10 from the base of the shoot. The leaf was enclosed in an aluminized plastic bag for at least one hour prior to measurement, following the methodology described by Scholander et al. [45], using a pressure chamber (Pump-Up Chamber, PMS Instrument Co., Ltd., Albany, OR, USA). Additionally, stomatal conductance (g_s_) was evaluated using a portable porometer (model SC1 from Decagon Devices) on fully expanded mature leaves. Temperature and relative humidity were recorded throughout the experiments using a hygrothermograph integrated into a data logger (HOBO^®^ Pro v2, Onset Computer Corporation, Bourne, MA, USA). These measurements served as the basis for estimating the vapor pressure deficit (VPD) both in the greenhouse and outdoors.

### 4.3. Plant Growth and Biomass Measurements

In both experiments, each week, the height of each plant was measured with a tape measure. Half of the plants were harvested after completing the 14-day treatment period (day 14), and the second half was harvested at the end of the recovery period (Rec). Fresh weight of roots, stems, and leaves was then determined, followed by dry weight measurement after 48 h in an oven at 70 °C.

### 4.4. RNA Extraction

In the scion/rootstock combination experiment, leaf samples were collected at the end of the stress period (day 14) and the subsequent recovery period (Rec) to evaluate the expression of foliar aquaporins. Leaf samples were collected from nodes 7 to 10 from the base of the shoot, always at the same time, between 09:00 and 11:00 h, to mitigate the circadian effect on aquaporin expression. Immediately after collection, the leaves were frozen in liquid nitrogen and stored at −80 °C until analysis.

For each experimental unit, 100 mg of tissue was ground with liquid nitrogen. Total RNA was extracted from three biological samples using the 3% cetyltrimethylammonium bromide (CTAB) protocol [46]. RNA concentration was measured using a NanoDrop 1000 spectrophotometer (Thermo Fisher Scientific Inc., Waltham, MA, USA), and its integrity was verified by electrophoresis. RNA was treated with RQ1 RNase-Free DNase (Promega) and used as a template to synthesize cDNA using Moloney murine leukemia virus reverse transcriptase (RT-MMLV; Promega). Each reaction mixture contained RNA template (2 μg), Oligo(dT)15 primers (2 μL), MMLV reverse transcriptase buffer 5× (5 μL), dNTPs (5 μL), recombinant ribonuclease inhibitor RNasin (0.63 μL), MMLV reverse transcriptase (1 μL), and DEPC-treated water (1.37 μL). The cDNA was diluted 1:4 (*v*/*v*) and a no-reverse transcriptase control was included.

Primer sequences for PIP and TIP aquaporins were selected based on previous studies by Gambetta et al. [47] and Zarrouk et al. [23]. To confirm the identity of the products, the primers were tested using RT-PCR, and the amplified products were sequenced. Primer sequences are provided in Table 1.

### 4.5. Quantitative PCR Analysis

Expression analysis was performed using real-time quantitative PCR (Stratagene Mx3000P). For transcriptional analysis, 1 μL of cDNA was used in the SYBR Green RT-PCR, along with 20 μL of Brilliant^®^ II SYBR^®^ Green qPCR Master Mix (Stratagene, Agilent Technologies Inc, Santa Clara, CA, USA) and 5 M of each primer in the thermocycler. The thermal profile for amplification included an initial denaturation step at 95 °C for 10 min, followed by 40 cycles at 95 °C for 30 s, 60 °C for 30 s, and 72 °C for 30 s. Additionally, a melting curve analysis was performed from 55 °C to 95 °C with 0.5 °C increments. Each reaction was run in duplicate and negative controls, including a water sample and a negative RT control, were included to ensure the absence of genomic DNA contamination. The specificity of the amplification products was confirmed by the presence of a single peak in the melting curve. Before analysis, the amplification efficiency of each primer pair was determined. The threshold cycle (Ct) values obtained were averaged across the two technical replicates for each sample and gene. Subsequently, the Ct values were normalized using the Ct values of the VvUBQ and VvAct genes, ensuring consistent gene expression variation following treatment application. The normalized Ct values were then used to evaluate changes

For each sample and gene, the threshold cycle (Ct) values obtained were averaged across the two technical replicates. Subsequently, the Ct values were normalized using the Ct values of the VvUBQ and VvAct genes, using the 2^−ΔΔCt^ method, ensuring consistent gene expression variation following treatment application. The normalized Ct values were then used to evaluate changes in expression levels [50]. The relative expression of the evaluated genes was calculated using the comparative Ct method (2^−ΔΔCt^) with three biological replicates and two technical replicates for each scion/rootstock combination.

### 4.6. Statistical Analysis

In both trials, a split-plot experimental design was used to evaluate the effects of irrigation treatments and genotypes or graft combinations on potted plants. The main plots were assigned to different irrigation levels (Fi or Ni), while the subplots focused on either genotypes or scion/rootstock combinations. For the genotypes experiment, eight whole plots were used, each randomized to one of the two irrigation levels, with four replications per level. Within each whole plot, two subplots were established, and one of four genotypes was randomly assigned to each subplot. In the combinations experiment, six whole plots were randomized to the two irrigation levels, with three subplots within each whole plot. Each subplot was then randomized to one of three scion/rootstock combinations. This design allowed for a comprehensive analysis of the interactions between irrigation treatments and plant genotypes or graft combinations. Data analysis was performed using linear mixed models according to the REML method in JMP pro software [51] (v.13). Differences between the means of irrigation treatments, genotype, or combinations were determined using Student’s *t*-test (*p* < 0.05) or the Tukey–Kramer test (*p* < 0.05). Regression coefficients were calculated using GraphPad Prism 9.3.1 (San Diego, CA, USA).

## 5. Conclusions

This study provides detailed insights into the varied responses of grapevine genotypes to water stress, emphasizing the significant role of rootstocks. In the first experiment, the non-grafted 103P rootstock demonstrated superior drought resistance by maintaining higher stomatal conductance and moderate stem water potential, indicative of an anisohydric response. In contrast, SO4 exhibited more pronounced stomatal closure, consistent with near-isohydric behavior. Meanwhile, CS and CAR displayed intermediate stomatal sensitivity. The second experiment highlighted the impact of irrigation treatments on growth rates, with CS/1103P and CS/101-14 combinations being the least affected, suggesting these rootstocks enhance drought resilience. The analysis of aquaporin expression revealed that TIP aquaporins, particularly *VvTIP2;1*, play a crucial role in water movement and recovery post-stress, with expression patterns closely linked to stomatal conductance. These findings underscore the importance of selecting appropriate rootstock/scion combinations and understanding their physiological mechanisms to improve drought resilience in grapevines. The study also highlights the need for refined methods to accurately characterize plant water use strategies, considering the dynamic nature of isohydric and anisohydric behaviors under varying environmental conditions.

## Figures and Tables

**Figure 1 plants-13-03427-f001:**
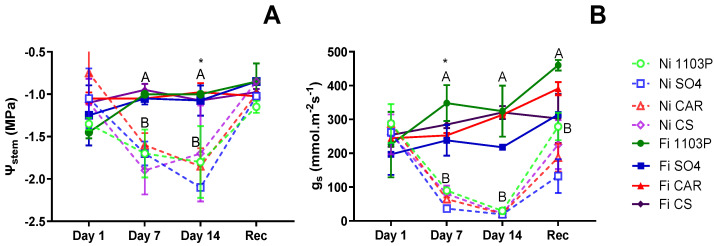
Stomatal conductance (g_s_) (**A**) and stem water potential (Ψ_stem_) (**B**) for own-rooted genotypes subjected to water stress treatment. Filled symbols with continuous lines correspond to irrigated plants, and empty symbols with a dashed line correspond to non-irrigated plants. Each point represents two sample media ± SE. Differences due to genotype are marked with an asterisk (*p* = 0.007). Capital letters indicate differences due to irrigation treatment (*p* < 0.001). The interaction of genotype × irrigation was not statistically significant (*p* = 0.261).

**Figure 2 plants-13-03427-f002:**
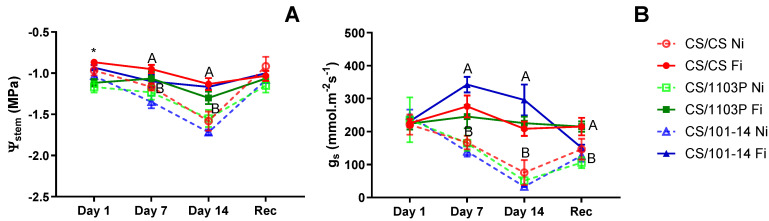
(**A**) Stem water potential (Ψ_stem_) and (**B**) stomatal conductance (g_s_) for scion/rootstock combinations subjected to water stress treatment. Filled symbols with continuous lines correspond to irrigated plants, and empty symbols with dashed lines correspond to non-irrigated plants. Each point represents three sample media ± SE. Differences due to genotype are marked with an asterisk (*p* = 0.00063). Capital letters indicate differences due to irrigation treatment (*p* < 0.001).

**Figure 3 plants-13-03427-f003:**
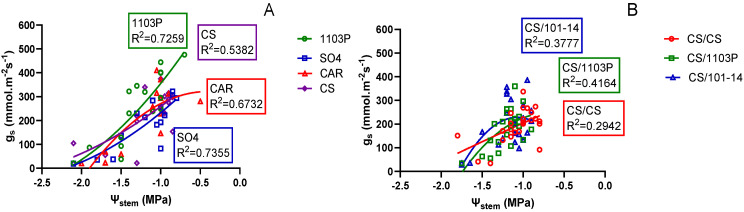
Relationship between stomatal conductance (g_s_, mmol m^−2^ s^-1^) and stem water potential (Ψ_stem_, MPa) in own-rooted genotypes (**A**) and scion/rootstock combinations (**B**). In (**A**), statistically significant differences from ANCOVA were found for genotypes (*p* = 0.0083) and Ψ_stem_ (*p* < 0.001). Each point represents one sample.

**Figure 4 plants-13-03427-f004:**
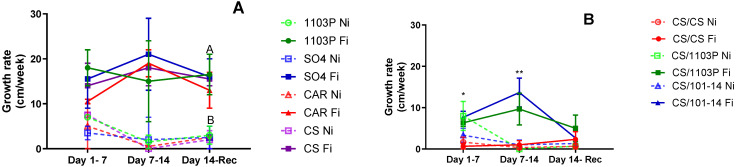
Shoot height growth rate for own-rooted genotypes (**A**); each point represents two sample media ± SE. Plant height growth rate for scion/rootstock combinations (**B**); each point represents three sample media ± SE. Filled symbols with continuous lines correspond to irrigated plants, and empty symbols with dashed lines correspond to non-irrigated plants. Capital letters indicate differences due to irrigation treatment. One asterisk represents differences due to combination, and two asterisks represent interaction between the irrigation and combination test.

**Figure 5 plants-13-03427-f005:**
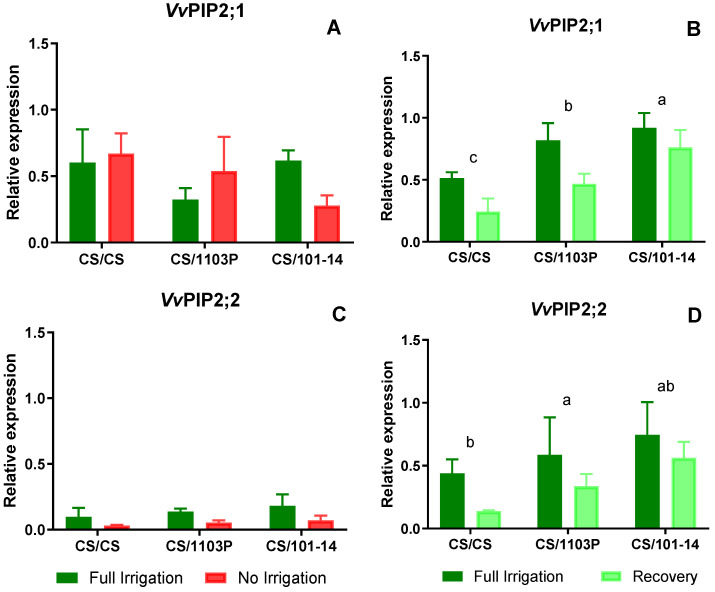
PIP aquaporin relative expression level (relative to *Vv*Ubiquitin and *Vv*Actin) in leaves of grafted plants after 14 days of treatment (**A**,**C**) and after one week in recovery (**B**,**D**). Bars represent three sample media ± SE. Different lowercase letters indicate significant differences between combinations determined by Student’s *t*-test.

**Figure 6 plants-13-03427-f006:**
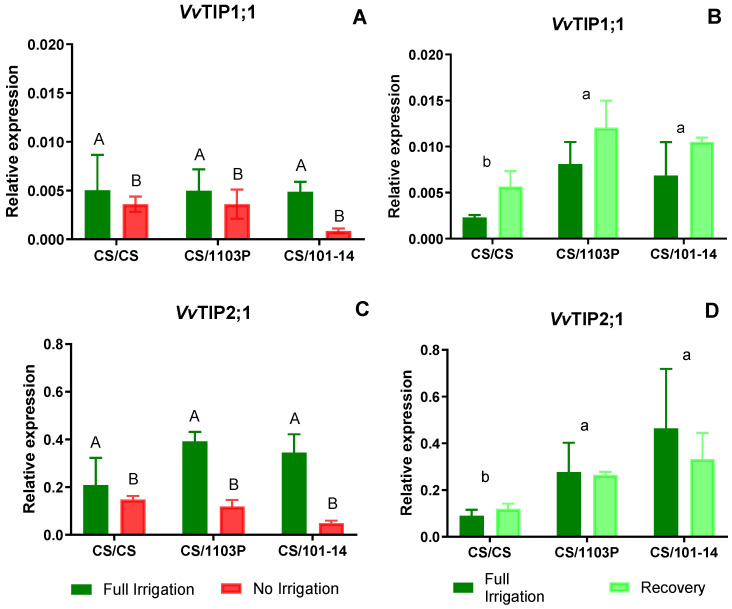
TIP aquaporin relative expression level (relative to *Vv*Ubiquitin and *Vv*Actin) in leaves of grafted plants after 14 days of treatment (**A**,**C**) and after one week in recovery (**B**,**D**). Bars represent three sample media ± SE. Capital letters indicate differences due to water treatment determined by Student’s *t*-test. Different lowercase letters indicate significant differences between combinations determined by the Tukey–Kramer test.

**Figure 7 plants-13-03427-f007:**
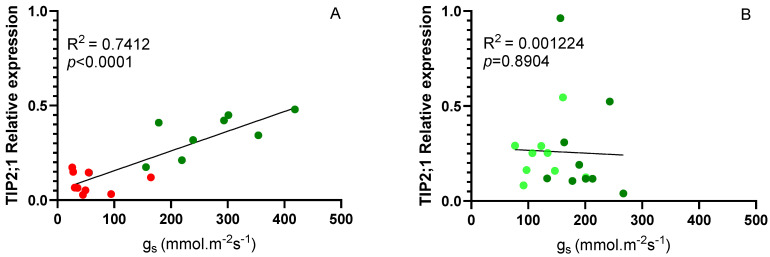
Correlation between the *VvTIP2;1* aquaporin expression level and stomatal conductance (g_s_) for the scion/rootstock combinations (**A**) at the end of the 14-day water stress period and (**B**) at the end of the recovery period. Each point represents one sample. Red circles represent water-stressed plants, dark green circles represent well-watered plants, and light green circles represent recovery plants.

**Table 1 plants-13-03427-t001:** References for aquaporin genes and sequences of primers used for quantitative PCR.

*Vitis* Gene	Fw/Rev	Sequence	Reference
*Vv* *PIP2;1*	Fw	5′-CCATTTTGATACCTTCTTCC-3′	Gambetta et al. [47]
	Rev	5′-TATCTACAATTTCATGCCCT-3′	
*VvPIP2;2*	Fw	5′-AACTAAAAACCCACAACACCC-3′	Gambetta et al. [47]
	Rev	5′-CATCATCATAATCATCTCTGG-3′	
*VvTIP1;1*	Fw	5′-AGCCTTTATTGGCGGACACA-3′	Zarrouk et al. [23]
	Rev	5′-GTAAACCAGGCCGAAGGTCA-3′	
*VvTIP2;1*	Fw	5′-AGGAGGAAGAGCAAGTTGTG-3′	Zarrouk et al. [23]
	Rev	5′-ACCAAAGCAAGGGCTTTACA-3′	
*Vv* *TIP2;2*	Fw	5′-CCATTGTTGCTTGCCTTCTCC-3′	Zarrouk et al. [23]
	Rev	5′-TTGGCACCCACTATGAACCC-3′	
*Vv* *Actin*	Fw	5′-CTTGCATCCCTCAGCACCTT-3′	Reid et al. [48]
	Rev	5′-TCCTGTGGACAATGGATGGA-3′	
*Vv* *Ubiquitin*	Fw	5′-TCTGAGGCTTCGTGGTGGTA-3′	Fujita et al. [49]
	Rev	5′-AGGCGTGCATAACATTTGCG-3′	

## Data Availability

The data presented in this study are available on request from the corresponding author.

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
