# Peer review of "Leaf Aquaporin Expression in Grafted Plants and the Influence of Genotypes and Scion/Rootstock Combinations on Stomatal Behavior in Grapevines Under Water Deficit"

_plants, 2024, doi:10.3390/plants13233427_

Round 1
Reviewer 1 Report
Comments and Suggestions for Authors
The manuscript attempts to evaluate the impact of various grapevine varieties and rootstock combinations on drought tolerance in Cabernet Sauvignon, employing both physiological and molecular analyses. However, the study has significant limitations and requires substantial revision. The detailed comments below outline specific areas for improvement.
1. The use of two differentiated experiments lacks coherence. For instance, in the first experiment, rootstock 101-14 was not assessed, with SO4 used instead. This inconsistency between the two experiments suggests that they may have been combined post hoc, rather than integrated into a unified, well-designed study.
2. The title should be revised to better reflect the actual findings. While the authors conclude that genotype influences aquaporin expression under water deficit, this aspect was not directly investigated in the study.
3. The study’s aim should be clarified and more specifically stated to provide a clearer direction for readers.
4. Lines 55-56. Please rewrite or better explain this assumption.
5. Lines 95-105. This paragraph is redundant with the Materials and Methods section; please remove.
6. Line 130. The term stem water potential is not well written (here and sometimes throughout the text).
7. Lines 161-162: Did you conduct a statistical test to support this assertion?
8. Line 170. Confirm if the intended word is "subdued."
9. Lines 281-282. Were less extreme than what?
10. The variation in Ψstem and gs between experiments is attributed to different environmental conditions, but could also stem from differences between single varieties versus grafted plants.
11. Lines 311-314. Consider referencing Labarga et al. 2023 (Plants, 12, 718) for additional insights.
12. Lines 321-322: If comparing with the NI period, please indicate that this comparison is statistically validated, as it currently lacks this analysis in the manuscript.
13. Line 388. The use of three-liter pots for rootstock studies is quite small and may not adequately represent rootstock effects on grafted plants. Additionally, leaving these plants unwatered for 14 days under high VPD raises credibility concerns. Were non-irrigated (NI) plants supplemented with water during this period to sustain them?
12. Line 418. There is here a sentence that should be removed.
13. Line 439. The leaf samples for aquaporin expression were collected at a different time from physiological measurements, potentially compromising comparability. Please justify this approach.
14. Lines 477-480. Please rewrite and better specify this part.
15. Please clarify how expression levels were calculated. Additionally, explain why RNA extraction was limited to leaves and why only aquaporin transcript levels were measured rather than including protein-level analysis.
16. The conclusions are too general.
Figures
Figure 1. It is unclear if the asterisks denote genotype differences only within NI plants or if they also include FI plants.
Figure 2. Standardize the symbology between Figure 2 and the aquaporin expression figures to avoid confusion (e.g., capital letters for genotype in one, treatment in another).
Figure 5. The legend mentions lowercase letters, which are absent from the figure; please remove or revise. Additionally, panel 5D does not align with its description in the text (line 210).
Figure 7. Adjust the green color shades to improve differentiation.
Comments on the Quality of English Language
Must be improved
Author Response
- The use of two differentiated experiments lacks coherence. For instance, in the first experiment, rootstock 101-14 was not assessed, with SO4 used instead. This inconsistency between the two experiments suggests that they may have been combined post hoc, rather than integrated into a unified, well-designed study.
-
The title should be revised to better reflect the actual findings. While the authors conclude that genotype influences aquaporin expression under water deficit, this aspect was not directly investigated in the study.
Response, regarding comments 1 and 2: The trials were carried out with a single objective, which was to advance the characterization of the grapevine’s response to water stress. Although it is true that there was no intention to compare the results of both trials, and that it was not possible to use the same rootstocks, we believe that both results are of interest for a better understanding of our subject. Therefore, we consider it appropriate to present them in the same paper. To incorporate this idea, we have changed the title to Leaf Aquaporine Expression in Grafted Plants, and Influence of Genotypes and Scion/Roostock Combinations in Stomatal Behavior in Grapevines under Water Deficit. We think this describes better our study and the results we obtained.
-
The study’s aim should be clarified and more specifically stated to provide a clearer direction for readers. Response 3: We have changed the wording of the objectives to make it clearer; see lines 82 to 93.
-
Lines 55-56. Please rewrite or better explain this assumption. Response 4: We rewrote line 54 for a clearer explanation.
-
Lines 95-105. This paragraph is redundant with the Materials and Methods section; please remove. Response 5: Although the paragraph is indeed redundant, given that the results are presented before Materials and Methods, we believe that providing a prior explanation of the methodology helps to understand the results better.
-
Line 130. The term stem water potential is not well written (here and sometimes throughout the text). Response 6. This was a compatibility error in the writing programs, and we have corrected it throughout the text.
-
Lines 161-162: Did you conduct a statistical test to support this assertion? Response 7 A covariance analysis was conducted, and indicating statistically significant difference among genotypes (P=0.0083), and a significant effect of Ψstem (P< 0.001); see lines 161 to 163, and lines 172 y 173.
-
Line 170. Confirm if the intended word is "subdued." Response 8: The word was changed in line 168 to “attenuated”.
-
. Lines 281-282. Were less extreme than what? Response 9: The text in lines 277 to 280 was changed to improve clarity.
-
The variation in Ψstem and gs between experiments is attributed to different environmental conditions, but could also stem from differences between single varieties versus grafted plants. Response 10: Indeed, this is an alternative possible explanation. This possibility was incorporated in the text in lines 281 and 282.
-
Lines 311-314. Consider referencing Labarga et al. 2023 (Plants, 12, 718) for additional insights. Response 11: This reference was incorporated in the text, in line 309 to 313.
-
Lines 321-322: If comparing with the NI period, please indicate that this comparison is statistically validated, as it currently lacks this analysis in the manuscript. Response 12: The statistical comparison among combinations was significant at the end of the Recovery period only. No comparison was performed with the Ni period. To clarify this, the reference to Figures 5 B and D, and Fig 6 B and D was specified in line 317 and 323.
-
Line 388. The use of three-liter pots for rootstock studies is quite small and may not adequately represent rootstock effects on grafted plants. Additionally, leaving these plants unwatered for 14 days under high VPD raises credibility concerns. Were non-irrigated (NI) plants supplemented with water during this period to sustain them? Response 13: No water was supplied to the plants during the 14 days of this treatment. Despite this, the plants responded well to the recovery period. This can be seen in the photographs of the trial included in the supplementary information (Fig S1 and Fig. S2), where the plants on day 14 show clear signs of dehydration, but there is no death.
- Line 418. There is here a sentence that should be removed. Response 14: The sentence was removed.
- Line 439. The leaf samples for aquaporin expression were collected at a different time from physiological measurements, potentially compromising comparability. Please justify this approach. Response:15. Leaf sample collection was carried out on the last day of irrigation treatments (days 14 and 21), to capture the differences triggered by the treatments. Since the irrigation treatments have been imposed well before sampling for gene expression and physiological measurements, we do not think that the comparability is being compromised by this protocol. Furthermore, in recognition that transcriptomic changes should occur in advance of any change in physiological traits, we followed this sequence of sampling that has been used in several previously published works (Vandeleur et al., 2009; Labarga et al., 2023, and others).
- Lines 477-480. Please rewrite and better specify this part. Response 16:
The text was modified to improve clarity and methodological details between lines 479 and 489.
-
Please clarify how expression levels were calculated. Additionally, explain why RNA extraction was limited to leaves and why only aquaporin transcript levels were measured rather than including protein-level analysis. Response 17: Expression levels were calculated using the Comparative Ct method (2-DDCt) and its description was included in the text. Although RNA extraction from root tips was initially attempted, the quality was insufficient for reliable analysis, leading us to focus on leaf samples where high-quality RNA was obtained. The study aimed to evaluate immediate transcriptional responses, best captured at the RNA level, hence protein-level analysis was not conducted as it would not directly reflect the transcriptional changes we sought to measure.
-
The conclusions are too general. Response 18. We changed the wording and content of the conclusions, and we added detail (see lines 499 to 514).
Figures
Figure 1. It is unclear if the asterisks denote genotype differences only within NI plants or if they also include FI plants.
Response: The legend of Figure 1 was modified to clarify that the interaction genotype x irrigation treatment was not significant and therefore the significant difference among genotypes indicated by the asterisk was obtained across irrigation treatments (Fi and Ni).
Figure 2. Standardize the symbology between Figure 2 and the aquaporin expression figures to avoid confusion (e.g., capital letters for genotype in one, treatment in another).
Response: Agreed, this change was done.
Figure 5. The legend mentions lowercase letters, which are absent from the figure; please remove or revise. Additionally, panel 5D does not align with its description in the text (line 210).
Response: The legend was fixed and the reference to lowercase letter was removed.
Figure 7. Adjust the green color shades to improve differentiation.
Response: Agreed, we improved the contrast of the green shades in Figure 7.
Reviewer 2 Report
Comments and Suggestions for Authors
Major revision is needed to improve this manuscript
Line 34-38: Incorporate names of the rootstocks involved in increasing drought tolerance, nutrient uptake, etc
Line 391: Provide recipe of 50% Hoagland solution.
Line 403: For how many days was this treatment applied?
I can’t understand treatments, varieties, or the number of plants used in 1st and 2nd experiment. To avoid confusion, it is advised to explain both experiments under individual headings.
Line 418: “Sure, here’s a clearer and improved version of the sentence: "What is meant by this line?
Line 429-434: Is this only for water deficit treatment? for both experiments?
Line 236-237: Why foliar aquaporin expression was not studied in experiment 1 (self-rooted genotypes) as depicted from manuscript title that aquaporin expression was studied both in genotypes in scion/stock combination
Above ground parameters mentioned in methodology were plant height, fresh weight of root, stem and leaves, and dry weight, but in results only shoot growth rate data was presented. Where are other parameters?
How control plants were irrigated and in scion/stock combination what control is used.
Line 233: “Considering the differences in the expression of VvTIP 2;1 aquaporins between genotypes (Fig 7 B)”
· This figure (7B) presents aquaporin expression for recovered plants while above statement stated that this figure is for genotypes (there is no data presented aquaporin expression in genotypes). Clarify this statement.
pictorial presentation is missing. Kindly add pictures of plant growth in different irrigation treatments
Comments on the Quality of English LanguageOverall, this manuscript is well-written and thoroughly described. However, there are many English sentence mistakes in the the overall write-up that cannot be overlooked at this stage. Therefore, some English mother tongue experts require editing of English language and style.
Author Response
- Line 34-38: Incorporate names of the rootstocks involved in increasing drought tolerance, nutrient uptake, etc. Response 1: In lines 34 to 38, we discuss the general influence of rootstocks on vigor, nutrient absorption capacity, and drought resistance. Specific examples of rootstocks with contrasting attributes in these areas are detailed in the cited references.
- Line 391: Provide recipe of 50% Hoagland solution. Response 2: We modified the text to mention the specific commercial source (brand and manufacturer) of the Hoagland solution used in the trails (lines 386 – 387).
- Line 403: For how many days was this treatment applied? Response 3: The text was edited to explain the duration of the treatment (21 days of full irrigation, see lines 398 to 404).
- I can’t understand treatments, varieties, or the number of plants used in 1stand 2nd To avoid confusion, it is advised to explain both experiments under individual headings. Response 4: We have changed the wording of the Experimental Design section (line 479 to 489) in order to make it clearer. We also rewrote the paragraph in line 398 to 404, where irrigation treatments Fi, Ni, and Rec are explained. The total number of plants is indicated on line 395.
- Line 418: “Sure, here’s a clearer and improved version of the sentence: "What is meant by this line? Response 5: We agree, the sentence was removed.
- Line 429-434: Is this only for water deficit treatment? for both experiments? Response 6: We rewrote line 425 for clearer explanation.
- Line 236-237: Why foliar aquaporin expression was not studied in experiment 1 (self-rooted genotypes) as depicted from manuscript title that aquaporin expression was studied both in genotypes in scion/stock combination. Response 7: Unfortunately, it was not possible to perform aquaporin analysis in experiment 1. However, we have changed the title to “Leaf Aquaporine Expression in Grafted Plants, and Influence of Genotypes and Scion/Roostock Combinations in Stomatal Behavior in Grapevines under Water Deficit”. We think this describes better our study and the results we obtained.
- Above ground parameters mentioned in methodology were plant height, fresh weight of root, stem and leaves, and dry weight, but in results only shoot growth rate data was presented. Where are other parameters? Response 8: Originally, these data were not included due to the lack of significant effects. New Supplementary Material was included to provide this information (Table S3 to Table S12).
- How control plants were irrigated and in scion/stock combination what control is used. Response 9: The text was modified to improve clarity and methodological details between lines 398 and 404.
- Line 233: “Considering the differences in the expression of VvTIP 2;1 aquaporins between genotypes (Fig 7 B)”. Response 10: We agree, and we rewrote lines 231 to 232.
- This figure (7B) presents aquaporin expression for recovered plants while above statement stated that this figure is for genotypes (there is no data presented aquaporin expression in genotypes). Clarify this statement. Response 11: Correct, Figure 7B corresponds to the expression of aquaporins in the scion/rootstock combinations test, this was fixed in line 241.
- Pictorial presentation is missing. Kindly add pictures of plant growth in different irrigation treatments. Response 12: In the Supplementary Information we add photos (Figure S1 and S2) of the plants of both treatments.
Reviewer 3 Report
Comments and Suggestions for Authors
Reviewers’ comments
The manuscript “Genotypes and scion/rootstock combinations influence stomatal behavior and aquaporin expression in grapevines under water deficit” offers a novel perspective, exploring the physiological responses of grapevines to water stress, particularly the role of rootstocks and aquaporins, which is an important theme closely related to grape cultivation and drought adaptation. In my opinion, some issues should be further improved in order to obtain a better and more informative work to be considered for publication.
1.Supplement the specific characteristics of the grape varieties and rootstocks used in the experiment."
2.Are there other growth indicators such as chlorophyll content, root system growth, photosynthesis rate, leaf size, and morphology to understand the impact of water on leaf and root growth?
3.In the references, some abbreviations and proper nouns need to be capitalized, and Latin names need to be in italics.
Author Response
- Supplement the specific characteristics of the grape varieties and rootstocks used in the experiment." Response 1: This information was added in the supplementary material Table S2.
- Are there other growth indicators such as chlorophyll content, root system growth, photosynthesis rate, leaf size, and morphology to understand the impact of water on leaf and root growth? Response 2: This information was added in the supplementary material Table S3 to Table S11, and Figure S1 and S2.
- In the references, some abbreviations and proper nouns need to be capitalized, and Latin names need to be in italics. Response 3: OK, we have revised the document and we corrected these errors.
Round 2
Reviewer 1 Report
Comments and Suggestions for Authors
I appreciate the thorough responses to the reviewers' comments. The article may be published in its current form.
Reviewer 2 Report
Comments and Suggestions for Authors
I appreciate the efforts put in by the authors to address all my concerns.
I feel that the MS can now be accepted in its present form.
Good Luck!
Reviewer 3 Report
Comments and Suggestions for Authors
The authors have supplemented the data to make the manuscript more in-depth. At the same time, the authors made detailed modifications to the References, including some details. Therefore, I believe that the manuscript version can be published.